# Vestibular Nuclei: A New Neural Stem Cell Niche?

**DOI:** 10.3390/cells11223598

**Published:** 2022-11-14

**Authors:** Guillaume Rastoldo, Isabelle Watabe, Agnes Lapotre, Alain Tonetto, Alejandra López-Juárez, Brahim Tighilet

**Affiliations:** 1Aix Marseille Université-CNRS, Laboratoire de Neurosciences Cognitives, LNC UMR 7291, 13331 Marseille, France; 2Fédération de Recherche Sciences Chimiques Marseille FR 1739, Pôle PRATIM, 13331 Marseille, France; 3Department of Chemical, Electronic and Biomedical Engineering, Division of Sciences and Engineering, University of Guanajuato, Guanajuato 38116, Mexico; 4GDR Physiopathologie Vestibulaire-Unité GDR2074 CNRS, 13331 Marseille, France

**Keywords:** stem cells, vestibular nuclei, neurogenesis

## Abstract

We previously reported adult reactive neurogliogenesis in the deafferented vestibular nuclei following unilateral vestibular neurectomy (UVN) in the feline and the rodent model. Recently, we demonstrated that UVN induced a significant increase in a population of cells colocalizing the transcription factor sex determining region Y-box 2 (SOX2) and the glial fibrillary acidic protein (GFAP) three days after the lesion in the deafferented medial vestibular nucleus. These two markers expressed on the same cell population could indicate the presence of lesion-reactive multipotent neural stem cells in the vestibular nuclei. The aim of our study was to provide insight into the potential neurogenic niche status of the vestibular nuclei in physiological conditions by using specific markers of stem cells (Nestin, SOX2, GFAP), cell proliferation (BrdU) and neuronal differentiation (NeuN). The present study confirmed the presence of quiescent and activated adult neural stem cells generating some new neurons in the vestibular nuclei of control rats. These unique features provide evidence that the vestibular nuclei represent a novel NSC site for the generation of neurons and/or glia in the adult rodent under physiological conditions.

## 1. Introduction

Neurogenesis is defined as the process leading to the genesis of functional neurons from neural stem cells (NSCs). The subventricular zone (SVZ) bordering the lateral ventricles and the subgranular zone (SGZ) within the dentate gyrus of the hippocampus are two primary neurogenic niches in the adult mammalian brain where endogenous neural stem cells (NSC) reside [1]. These NSCs derive from a single cell type, the germinal astrocyte, which can be identified by specific nuclear factors (Sox1, Sox2) and intermediate filament proteins (nestin, glial fibrillary acidic protein [GFAP] and vimentin) [2,3] (for review, see [4,5]). While spontaneous adult neurogenesis, under physiological conditions, produces mature neurons that migrate, differentiate and then functionally integrate into preexisting neural networks, different pathologies or lesions of the central nervous system can stimulate neurogenesis in the classical niches (SVZ and SGZ) but also in other brain regions [6,7]. This process is known as neurogenesis in pathological conditions, or reactive neurogenesis. Indeed, under the influence of specific signals, a tissue previously considered as non-neurogenic can overcome endogenous restrictions and allow a time-limited neurogenesis in the damaged area. Two possibilities exist to explain the neurogenesis observed in an affected brain region: either the structure hosts NSCs that will be able to produce new neurons, or the new neurons come from the classical niches (mainly SVZ) through different pathways (along blood vessels or via cerebrospinal fluid) [8,9].

Several notable facts led us to study the neurogenic status of the vestibular nuclei. First, our previous studies demonstrated reactive neurogenesis in the vestibular nuclei in the feline [10,11,12,13,14] and in the rat [15,16,17] following unilateral vestibular neurectomy (UVN). Then, on the rodent model, we identified multiple SOX2+ cells (one of the markers of germinal astrocytes) within the vestibular parenchyma, as well as a drastic increase of these cells a few days after UVN [16].

The aim of the present study was to demonstrate that proliferating cells, found in the vestibular nuclei of control animals, express proteins typically associated with NSCs. We also aimed to demonstrate that these cells give rise to neurons in vivo without inducing vestibular nerve damage. These unique features provide evidence that the vestibular nuclei represent a novel NSC site for the generation of neurons and/or glia in the adult rodent in physiological conditions.

## 2. Materials and Methods

### 2.1. Animals

The experiments were performed on 4 adult Long Evans rats (250/300g) originating from Charles River (St Germain sur l’Arbresle, France). All experiments were performed in accordance with the National Institutes of Health Guide for Care and Use of Laboratory Animals (NIH Publication no. 80-23) revised in 1996 for the UK Animals (Scientific Procedures) Act of 1986 and associated guidelines or the Policy on Ethics approved by the Society for Neuroscience in November 1989 and amended in November 1993, and under the veterinary supervision and control of the National Ethical Committee (French Agriculture Ministry Authorization: B13-055-25). The present study was specifically approved by Neurosciences Ethic Committee N°71 of the French National Committee of animal experimentation. Every attempt was made to minimize both the number and the suffering of animals used in this experiment. The animals were housed in a large confined space with 12h-12h diurnal light variations with free access to food and water.

### 2.2. Tissue Preparation

The rats were deeply anesthetized with a mixture of ketamine 1000/medetomidine and then perfused by intracardiac injection. The intracardiac injection of 400 mL of isotonic saline (0.9% NaCl) was followed by an infusion of 400 mL of freshly prepared Zamboni solution (4% paraformaldehyde and in 0.1 M phosphate buffer (PB), pH 7.4). At the end of the perfusion, the brain was extracted from the skull and post-fixed overnight at 4 °C in the same fixative mixture as that used during the perfusion. Brains were rinsed and cryoprotected by successive transfers into increasing concentrations (10%, 20%, 30% of sucrose solution in 0.1 M PB for 72 h at 4 °C). Brains were rapidly frozen with CO_2_ gas and cut into serial 40 µm frontal sections with a cryostat (Leica) for immunochemistry. Sections were stored at −20 °C in cryoprotectant solution until used.

### 2.3. Immunohistochemistry

Animals received 2 injections of BrdU (6 h apart; 50 mg/kg body weight, i.p) for three consecutive days and were perfused three weeks later to study neurogenesis in the adult rodent vestibular nuclei. For BrdU immunostaining, floating brain sections were washed (3 × 10 min) with PBS in multiwell plates and blocking was performed by incubation (2 h) in DMEM with 10% FBS. Sections were incubated with 2 N HCl and 0.5% Triton-X100 in PBS (30 min, 37 °C) for DNA hydrolysis. Then, sections were rinsed in 0.1 M sodium tetraborate buffer, pH 8.5, before overnight incubation with the primary antibody at 4 °C (rat anti-BrdU, 1:100 Oxford Biotech), followed by incubation with the secondary antibody for 2 h at room temperature. For all other antibodies, blocking was performed by incubation (1 h) in 5% BSA and 0.3% Triton X-100. Slides were incubated overnight at 4 °C with the primary antibody. To evaluate the neurogenic potential of the vestibular nuclei, we used different antibodies: goat anti-SOX2 (1:100, R&D Systems, Cat#AF2018) as a marker of multipotent stem cells [18] in colocalization with rabbit anti-GFAP (1:200, Dako, Santa Clara, CA, USA) or mouse anti-Nestin (1:100, R&D, Cat#MAB1259) that is known as a neural progenitor cell marker [19]. Fluorescent secondary antibodies (Abcam, Cambridge, UK) were used as follow: Alexa Fluor 594 nm goat anti-rat (1:500), Alexa Fluor 594 nm donkey anti-goat (1:500), Alexa Fluor 488 nm donkey anti-rabbit (1:500) and Alexa Fluor 488 nm donkey anti-mouse (1:500) for 2 h at room temperature. Finally, brain sections were mounted onto SuperFrost/Plus glass slides (Fischer, Innviertel, Austria) and air-dried before being mounted with Roti^®^ Mount FluorCare antifade reagent with the nuclear marker DAPI (Carl Roth, Karlsruhe, Germany).

### 2.4. Cells Count and Statistical Analysis

For quantification of cells expressing specific markers, 1 among 12 serial sections starting at the beginning of vestibular nuclei (relative to bregma, −9.84 mm) to the end of vestibular nuclei (relative to bregma, −13.08 mm), according to the rat brain stereotaxic atlas [20], were used. Only sections of the medial vestibular nucleus (MVN) were evaluated for cell counting. Quantification of SOX2-positive cells was performed using confocal imaging with a Zeiss LM 710 NLO laser scanning microscope equipped with a 63×/1.32 NA oil immersion lens. Immunoreactive positive cells in the MVN were counted using an integrated microscopic counting chamber that delineated the region of interest by a square of 425.10 μm^2^. The average cell counts from the sections were used for statistical analysis.

### 2.5. Statistical Analysis

We performed all statistical analyses using GraphPad Prism software (version 9.3.1, GraphPad Software, Harvey Motulsky, San Diego, CA, USA). Summary graphs were all shown as mean ± SEM. Statistical analyses were evaluated by one-way ANOVA. ANOVA was followed by post hoc analysis with the Tuckey’s test. *p*-value < 0.05 was considered as statistically significant. * *p* < 0.05, ** *p* < 0.01, *** *p* < 0.001.

## 3. Results

To assess the neurogenic potential of the vestibular nuclei in adult control animals, we analyzed the expression of a multipotent stem cell marker, SOX2, and spontaneous cellular proliferation with the proliferative marker 5-bromodeoxyuridine (BrdU). We detected SOX2-positive (+) and BrdU+ cells inside classical neurogenic regions, such as the olfactory bulb (OB), subventricular zone (SVZ), subgranular zone (SGZ), and the medial vestibular nuclei (MVN) (Figure 1a). Quantification inside classical neural stem cell (NSC) niches revealed that the OB and the SVZ contained more SOX2+ cells compared to the MVN (*p* < 0.0001 for both OB and SVZ; Figure 1b). However, similar values of SOX2+ cells were found in the SGZ and the MVN (*p* > 0.05; Figure 1b). In brain sections at the vestibular nuclei level, this cell population appears to be restricted to the vestibular nuclei, as illustrated in Figure 1c. To confirm the pluripotency of SOX2+ cells observed in the MVN of adult control animals, we used two different cellular markers: GFAP and Nestin. We found that more than 90% of SOX2+ cells co-expressed GFAP (Figure 1d,e, Z-stack and 3D reconstruction are presented in Appendix A) and also expressed Nestin (Figure 1f), confirming the presence of quiescent and proliferating NSCs (SOX2+/BrdU+) in the MVN.

Interestingly, few proliferative cells (BrdU+) were found in all vestibular nuclei (inferior, medial, lateral and superior nuclei) of the adult control rat (Figure 2a). To confirm the presence of neurogenesis endemic to the vestibular nuclei in physiological conditions, we used a double labeling with BrdU and NeuN. Thus, in the present study, we demonstrate the formation of new neurons (BrdU+/NeuN+) within the vestibular nuclei of adult rats (Figure 2b, Z-stack and 3D reconstruction are presented in Appendix A). Taken together, these results confirmed the presence of quiescent and activated NSCs generating new neurons in the vestibular nuclei of adult control animals.

## 4. Discussion

In adult rodents, NSCs have been widely documented in the SVZ and the SGZ [21,22], but the hypothalamus, the neocortex and the inferior colliculus have recently been identified as new potential neurogenic zones [23,24,25,26]. In these neurogenic regions, quiescent and activated NSCs are present, generating new neurons throughout life [25,26,27]. To highlight the potential neurogenic niche status of the vestibular nuclei in physiological conditions, we used a combination of cell markers to identify NSCs. We demonstrated in vivo the presence of both SOX2+/GFAP+ and SOX2+/Nestin+ cells in the parenchyma of the MVN of intact animals, suggesting a pool of quiescent NSCs in the vestibular nuclei. These results corroborate previous observations of multipotent NSCs in the mammal brainstem [28]. In addition, we also found sparse proliferative cells (BrdU+) in all vestibular nuclei of intact rats. More specifically, colocalization of SOX2 and BrdU labeling indicated that in control rats, the MVN, the olfactory bulb, the SVZ and the SGZ contain quiescent and activated NSCs. Finally, we demonstrated an endogenous neurogenesis (BrdU+/NeuN+) in physiological conditions in the vestibular nuclei. Based on all this evidence, we may wonder whether the vestibular nuclei can be considered as a NSC niche, as is the case for the SVZ and the SGZ. By synthesizing data from several studies (for review, see [22,29]), we are able to propose a list of criteria that would assign a NSC niche status to a particular cerebral region. Obviously, not all niches integrate all these distinct elements, and it is clear that a neurogenic niche represents a complex and dynamic entity in which the integration of multiple signals allows precise control of the NSCs proliferation and survival.

First, recent studies suggest that the extracellular matrix (ECM) is an essential component of NSC niches and regulates adult neurogenesis [30,31]. Specifically, chondroitin sulfate proteoglycans and hyaluronan influence NSCs in the SVZ and SGZ. These two components of the ECM are highly expressed in the vestibular nuclei of adult rats [32] and their expression is modified during vestibular compensation [33]. In addition, the reorganization of ECM components within the vestibular nuclei returns to control values at the end of the first week, which is temporally correlated with both the restoration of electrophysiological homeostasis in the vestibular nuclei (VN) and a recovery of vestibular functions [34]. Furthermore, extensions of EMC in the SVZ, called fractones, play a central role in adult NSC niches [31]. Fractones can contact many NSCs and other cells in the ventricle and can specifically sequester growth factors to modulate cell proliferation in the SVZ [35,36]. Interestingly, this key structure of ECM in the SVZ is also present along the walls of the fourth ventricle [37], adjacent to the MVN. In addition, the anatomical proximity of the vestibular nuclei to the circumventricular organs, structures that line the 3rd and 4th ventricles, also completes the list of criteria for a NSC niche.

Another specificity of NSC niches is the presence of a vascular niche within them. Vascularization in adult SVZ and SGZ is very dense and highly organized compared to non-neurogenic brain regions [38]. To this end, it is worth noting that among all structures of the rat brainstem, the medial vestibular nucleus is the most vulnerable to ischemia, which attests to an important vascularization of this nucleus [39].

There are specificities peculiar to the SVZ or SGZ niche; therefore, it is not surprising that there are particularities within the vestibular nuclei that differentiate them from the classical niches. As an example, despite the presence of many quiescent NSCs in the vestibular nuclei, almost equivalent to those found in the SGZ, physiological neurogenesis in this region remains weak compared to classical niches. Thus, in spite of the multiple elements that could position the vestibular nuclei as a neurogenic niche, there are still some grey areas. What is the physiological interest of this important pool of quiescent NSCs in the vestibular nuclei? Indeed, we were the first to demonstrate the existence of a reactive neurogenesis restricted to the deafferented vestibular nuclei after UVN in the feline model [14]. We have also shown that a pharmacological blockade of this reactive neurogenesis delays the compensation of postural and locomotor functions, but not of oculomotor function (compensation of horizontal nystagmus) [10]. What are the precise cellular characteristics of these NSCs and what are the cellular signals that activate them? By comparing different animal models of vestibular lesions reproducing a reversible blocking of the peripheral vestibular inputs (tetrodotoxin injection in the inner ear), or a definitive loss of peripheral vestibular sensors (unilateral labyrinthectomy model), or a complete and irreversible ablation of the vestibular peripheral sensors and Scarpa’s ganglion (UVN model), we were able to demonstrate that vestibular reactive neurogenesis was strongly expressed only in the UVN model [11]. It would then appear that the abrupt section of the vestibular nerve induces a permissive environment, allowing the expression of important neurogenesis in the vestibular nuclei to compensate for the posturo-locomotor deficits in the feline model. In this regard, we have recently documented the conditions that allow the deafferented vestibular environment to express reactive neurogenesis. The strong glial and inflammatory reaction in the deafferented VN, generated by the section of the vestibular nerve seems to represent a key condition for this neuroplasticity process [40]. One can imagine the adaptive role of vestibular neurogenesis in the vestibulo-injured animal, but what can be its functional role in physiological conditions? Does it contribute to compensating for cell loss linked to aging of the vestibular system? Future studies should help to further extend this discovery and to better characterize vestibular neural stem cells to deepen the understanding of this physiological neurogenesis in the adult vestibular nuclei.

## Figures and Tables

**Figure 1 cells-11-03598-f001:**
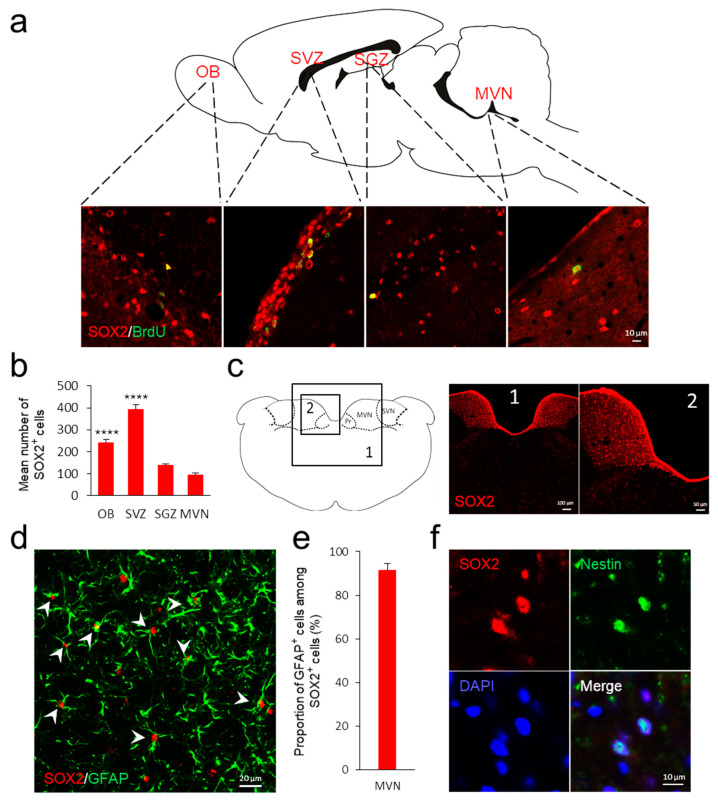
**Neural stem cells in classical niches and in the vestibular nuclei of adult rats**. (**a**) Schematic drawing of adult neurogenic niches of the rodent brain (OB: olfactory bulb, SVZ: subventricular zone, SGZ: subgranular zone, MVN medial vestibular nucleus). Lower part: confocal immunostaining images of newly generated (BrdU+, in green) and SOX2+ (in red) cells in coronal sections of OB, SVZ, SGZ and MVN of adult control rats. Scale bar, 10 μM. (**b**) Histograms comparing the number of SOX2+ cells in the OB, SVZ, SGZ and MVN of adult control rats. Error bars represent SEM. Statistical analysis were made only with the MVN. **** *p* < 0.0001. (**c**) SOX2+ cells in the vestibular nuclei. Left panel: schematic representation of a coronal section of the rat brainstem at the level of the vestibular nuclei. Right panel: presence of SOX2+ cells in the brainstem of an adult rat at the Interaural −2.64 mm and Bregma 11.64 mm level of the Paxinos and Watson stereotactic atlas. Scale applies to panel 1: 100 μM; panel 2: 50 μM. MVN = medial vestibular nucleus; SVN = superior vestibular nucleus; Pr = prepositus nucleus. (**d**) Merged representative image of MVN from adult control rats immunostained with SOX2 (red) and GFAP (green) antibody to identify neural stem cells. Arrowheads represent cells colocalizing SOX2 and GFAP marker. Scale bar, 20 μM. (**e**) Percentage of SOX2+ cells that colocalized with GFAP marker. (**f**) Representative MVN image from adult control rats stained with antibodies against SOX2, Nestin and DAPI to identify neural stem cells. Scale bar, 10 μM.

**Figure 2 cells-11-03598-f002:**
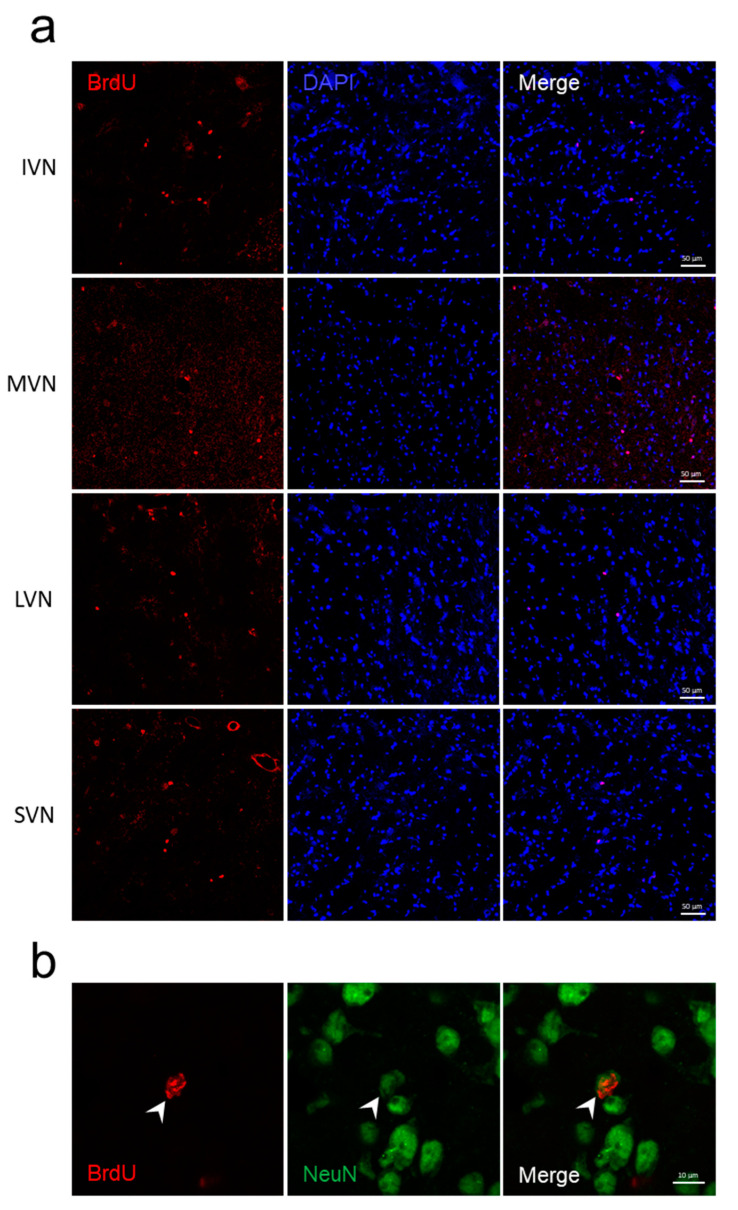
**Cell proliferation and differentiation in neurons in the vestibular nuclei of adult control rats**. (**a**) Representative images of proliferative cells (BrdU+) in all vestibular nuclei of adult control rats. IVN = inferior vestibular nucleus; MVN = medial vestibular nucleus; LVN = lateral vestibular nucleus; SVN = superior vestibular nucleus. Scale bar, 50 μM. (**b**) Maximum intensity orthogonal projection of the confocal Z-stack of newly generated neurons in the MVN of adult control rats. Arrowheads indicate the colocalization of BrdU+ (in red) and NeuN+ (neuronal marker in green). Scale bar, 10 μM.

## Data Availability

The data presented in this study are available on request from the corresponding author.

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
