# Peer review of "Vestibular Nuclei: A New Neural Stem Cell Niche?"

_cells, 2022, doi:10.3390/cells11223598_

Round 1
Reviewer 1 Report
This manuscript brings in situ demonstration for the existence of a new neural stem niche in the medial vestibular nucleus of rodents, via immunohistofluorescent labeling of currently established markers and confocal microscopy. This study provides a basic “missing gap” in the previous multistep detailed demonstration of a lesion-induced neurogenesis in the adult brain in the vestibular nuclei, first in cat and later in rat.
The experimental study is rigorously conducted, including positive controls on classical neurogenic niches and labeling quantifications. The results are fully demonstrative, the figures are fine, and the interpretations are straightforward and rigorous. The discussion is strongly argumented and opens original and up-to-date perspectives about the physiological meanings of adult neurogenesis, specially with regard to post-lesional reactions and repair. The whole paper is remarkably written and follows a linear and rigorous logic. The manuscript although requires some minor corrections in order be published.
Scientific issues
Page 1 (abstract), line 21: Replace “multipotent cells” by “lesion-reactive multipotent neural stem cells”.
Minor comments
Page 2, line 85: Update spelling of “[frontal] sections”.
P.2, l.86: Change “needed” into “used”.
P.2, l.89: Replace “after” by “later”.
P.2, l.9598: Add the bibliographic references “Ahlfeld et al 2017” and “Suzuki et al 2010” into the “References” section of manuscript, replace in text by numerical referencing, and update accordingly the references numbers in whole manuscript.
P.3, l.105: Change “1 in 12” into “1 among 12”.
P.3, l.106: add “the” ahead of “beginning”.
P.3, l.109: Change “counted” into “performed”.
P.6, l.209: Change “suggests” into “suggest”.
p.7, l.265: Correct “Recherche".
Author Response
We would like to thank the reviewer for his careful evaluation of our manuscript and for his positive statement. All suggested corrections have been made and included in the revised manuscript.
Reviewer 2 Report
The manuscript by Rastoldo et al on Vestibular nuclei: A new neural stem cells niche? focused majorly on characterization of neural stem cells isolated from vestibular nuclei. Authors characterized the neural stem cells by examine the neurogenic potential and cell proliferation capacity of NSCs localized in vestibular nuclei. However, the following concerns needs to be addressed.
Major comments:
1. Authors should perform additional methods to characterize neural stem cells. (For review PUBMED: 34662661, 29035086)
Author Response
The authors are grateful to the reviewer for giving us the opportunity to revise and improve the paper while offering his/her helpful and constructive criticism.
- We thank the reviewer for his remark to include other methods to better characterize vestibular neural stem cells. The purpose of this paper is to highlight the in-vivo presence of neural stem cells in the vestibular nuclei of adult rats. The different papers proposed isolated, cultured and finely characterized neural stem cells from canine fetal spinal cord (Pubmed: 34662661) or from human umbilical cord blood-derived mesenchymal stem cells (Pubmed: 29035086). Exactly as described in these two papers, it is important to perform a more detailed characterization in-vitro of vestibular stem cells. However, here we report neurogenesis and neural stem cells present in adulthood, thus in a mature brain, unlike the fetal cells used in the cited papers and the difference is also in the in-vivo condition of our study. Currently, with the lack of expertise in the laboratory and the isolation of adult neural stem cells being more complex to perform than for developing stem cells, we cannot better characterize vestibular neural stem cells and we have opted to only demonstrate the presence of these cells in-vivo. However, we have added a sentence at the end of the discussion supporting a better characterization of these cells and our results open the door to other studies to better understand adult vestibular neurogenesis.
Reviewer 3 Report
The authors demonstrated the presence of neural stem cells in the vestibular nucleus under physiological conditions using cell markers such as Sox2, Nestin, GFAP, and the cell cycle marker BrdU. The authors have long studied neurogenesis in the vestibular nucleus after unilateral vestibular nerve transection, and this study focused specifically on it under physiological conditions.
General remark;
The present study is supposed to provide basic data, but it lacks the more quantitative and methodologically sound experimental results that would be needed.
Specific comments;.
1) The authors use double staining of markers to demonstrate neural stem cells, but in laser confocal microscopy, the Z-stack images can mistake true double-stained cells. For example, Fig. 1d shows double staining of Sox2 (nuclear staining) and GFAP (cytoplasmic and cell process localization), but in our experience, it is dangerous to say that double staining is present only in images at this magnification and level. Only by using a 3D image (XZ and YZ images in addition to the XY image) at a higher magnification can we be sure that the image is double-stained.
(2) The authors quantified the number of Sox2-positive cells only in the MVN (Fig. 1b), but Fig. 2 shows BrdU-positive cells in each subnucleus of the vestibular nucleus. If the authors propose neurogenesis under physiological conditions in the entire vestibular nucleus, they should identify proliferating cells by long-term BrdU administration and provide quantitative data on the neurogenesis using double staining with Sox2.
3) The method of BrdU labeling and identification is questionable: BrdU detection requires denaturation of sections with acid, a process that usually results in the absence of DAPI staining. Since the Materials and Methods does not describe the BrdU staining method, this issue is difficult to evaluate and should be clarified by the authors.
(4) Aside from the double staining of BrdU and DAPI, double staining of NeuN and BrdU is shown in Fig. 2b. The cell indicated by the arrowhead is an example of coexistence of BrdU and NeuN, but this coexistence should also be shown as a 3D image by laser confocal microscopy to determine if it is a true coexistence. The strange thing is that the NeuN-positive structures are basically considered to represent cell nuclei, but the DAPI staining is very weak or absent in this figure. There are also fibrous and dotted NeuN-positive structures in this figure. I am wondering what these structures indicate.
Author Response
The authors are grateful to the reviewers for giving us the opportunity to revise and improve the paper while offering their helpful and constructive criticism.
1) The authors use double staining of markers to demonstrate neural stem cells, but in laser confocal microscopy, the Z-stack images can mistake true double-stained cells. For example, Fig. 1d shows double staining of Sox2 (nuclear staining) and GFAP (cytoplasmic and cell process localization), but in our experience, it is dangerous to say that double staining is present only in images at this magnification and level. Only by using a 3D image (XZ and YZ images in addition to the XY image) at a higher magnification can we be sure that the image is double-stained.
Response (1) We totally agree with this pertinent remark. To answer this comment we have made in Supplementary Figure 1 (a,b,c) orthogonal projection of confocal Z-stacks (a) with a 3D reconstruction (b) and single Z-slice (c) of the double staining SOX2/GFAP in order to confirm the colocalization of these two markers in the medial vestibular nucleus of adult rats. As you can see on Supplementary Figure 1b and 1c, GFAP cellular labeling clearly envelops the SOX2 positive nucleus. The total Z-stack thickness of 21µM ensures that all planes of SOX2/GFAP labelling are present and clearly confirms the colocalization of these two markers.
(2) The authors quantified the number of Sox2-positive cells only in the MVN (Fig. 1b), but Fig. 2 shows BrdU-positive cells in each subnucleus of the vestibular nucleus. If the authors propose neurogenesis under physiological conditions in the entire vestibular nucleus, they should identify proliferating cells by long-term BrdU administration and provide quantitative data on the neurogenesis using double staining with Sox2.
Response (2) We understand the concern, however, the purpose of this paper is to highlight the presence in-vivo of neural stem cells and the presence of neurogenesis in the vestibular nuclei of adult rats. Of course, it would be interesting in the long term to better characterize this adult neurogenesis in each vestibular nucleus and its functional implication. However, we think it is better to carry out this study in another paper and keep in this present paper the novelty of the discovery of a new adult neurogenic niche in the rat brain.
3) The method of BrdU labeling and identification is questionable: BrdU detection requires denaturation of sections with acid, a process that usually results in the absence of DAPI staining. Since the Materials and Methods does not describe the BrdU staining method, this issue is difficult to evaluate and should be clarified by the authors.
Response (3) We thank the reviewer for this comment and we modified the Material and Methods section 2.3 of the manuscript to describe precisely the BrdU staining method. Briefly, floating brain sections were washed (3x10 min) with PBS in multiwell plates and blocking was done by incubation (2h) in DMEM with 10% FBS. Sections were incubated with 2N HCl and 0.5% Triton-X100 in PBS (30min, 37°C) for DNA hydrolysis. Then, sections were rinsed in 0.1M sodium tetraborate buffer, pH 8.5, before overnight incubation with the primary antibody at 4°C followed by incubation with the secondary antibody for 2h at room temperature.
Finally, to answer the question about DAPI labelling, we suppose that we can see both BrdU and DAPI staining despite the denaturation of sections because we used Roti Mount Fluorcare Dapi Mounting Media in our protocol. However, to be totally transparent, it is true that adding 2N HCl during the immunohistochemical protocol damages the nervous tissue and we lose fluorescence on some nuclei contrary to an immuno with a classical protocol without acid.
(4) Aside from the double staining of BrdU and DAPI, double staining of NeuN and BrdU is shown in Fig. 2b. The cell indicated by the arrowhead is an example of coexistence of BrdU and NeuN, but this coexistence should also be shown as a 3D image by laser confocal microscopy to determine if it is a true coexistence. The strange thing is that the NeuN-positive structures are basically considered to represent cell nuclei, but the DAPI staining is very weak or absent in this figure. There are also fibrous and dotted NeuN-positive structures in this figure. I am wondering what these structures indicate.
Response (4) To answer this comment we have made in Supplementary Figure 1 (d,e,f) orthogonal projection of confocal Z-stacks (d) with a 3D reconstruction (e) and a single Z-slice (f) of the double staining BrdU/NeuN in order to confirm the colocalization of these two markers in the medial vestibular nucleus of adult rats. As you can see on Supplementary Figure 1e and 1f, BrdU nuclear labeling clearly colocalize with the NeuN positive cell. The total Z-stack thickness of 8µM ensures that all planes of BrdU/NeuN labelling are present and makes it impossible to label a different BrdU nucleus with the same neuron labeled with NeuN. These images clearly confirms the colocalization of these two markers.
Concerning the fibrous and dotted NeuN-positive structures showed in previous Figure 2B, we believe that this is a saturation/brightness default of the x63 zoom that we used to create the image. However, since the new BrdU/NeuN image presented in Supplementary Figure 1 is now better and shows less fibrous and dotted NeuN-positive structure, we replaced the Figure 2B.
Round 2
Reviewer 2 Report
Accept
Reviewer 3 Report
The authors responded to most of the comments I made. However, the authors refused to add the quantitation of stem cell-derived neurons in the subnuclei of the vestibular nucleus. Actually, the evidence of neurogenesis (i.e., coexistence of BrdU labeling and NeuN immunoreactivity) in the vestibular nucleus is limited to the image of a single cell (Revised Figure 2b, Supplementary figure also indicates the same cell). This dearth of quantitative analysis is the critical flaw of the present study.